# Application Potential of *Trichoderma* in the Degradation of Phenolic Acid-Modified Chitosan

**DOI:** 10.3390/foods12193669

**Published:** 2023-10-05

**Authors:** Maria Swiontek Brzezinska, Beata Kaczmarek-Szczepańska, Grażyna B. Dąbrowska, Marta Michalska-Sionkowska, Katarzyna Dembińska, Agnieszka Richert, Marcela Pejchalová, Sweta Binod Kumar, Agnieszka Kalwasińska

**Affiliations:** 1Department of Environmental Microbiology and Biotechnology, Faculty of Biological and Veterinary Sciences, Nicolaus Copernicus University in Torun, Lwowska 1, 87-100 Torun, Poland; mms@umk.pl (M.M.-S.); katarzynadembinska6@gmail.com (K.D.); sweta14789@gmail.com (S.B.K.); kala@umk.pl (A.K.); 2Department of Biomaterials and Cosmetics Chemistry, Faculty of Chemistry, Nicolaus Copernicus University in Torun, Gagarina 7, 87-100 Torun, Poland; 3Department of Genetics, Faculty of Biological and Veterinary Sciences, Nicolaus Copernicus University in Torun, Lwowska 1, 87-100 Torun, Poland; browsk@umk.pl (G.B.D.); a.richert@umk.pl (A.R.); 4Department of Biological and Biochemical Sciences, Faculty of Chemical Technology, University of Pardubice, Sudentska 573, 53210 Pardubice, Czech Republic; marcela.pejchalova@upce.cz

**Keywords:** chitosan, *Trichoderma*, biodegradation

## Abstract

The aim of the study was to determine the potential use of fungi of the genus *Trichoderma* for the degradation of phenolic acid-modified chitosan in compost. At the same time, the enzymatic activity in the compost was checked after the application of a preparation containing a suspension of the fungi *Trichoderma* (spores concentration 10^5^/mL). The *Trichoderma* strains were characterized by high lipase and aminopeptidase activity, chitinase, and β-1,3-glucanases. *T. atroviride* TN1 and *T. citrinoviride* TN3 metabolized the modified chitosan films best. Biodegradation of modified chitosan films by native microorganisms in the compost was significantly less effective than after the application of a formulation composed of *Trichoderma* TN1 and TN3. Bioaugmentation with a *Trichoderma* preparation had a significant effect on the activity of all enzymes in the compost. The highest oxygen consumption in the presence of chitosan with tannic acid film was found after the application of the consortium of these strains (861 mg O_2_/kg after 21 days of incubation). Similarly, chitosan with gallic acid and chitosan with ferulic acid were found after the application of the consortium of these strains (849 mgO_2_/kg and 725 mg O_2_/kg after 21 days of incubation). The use of the *Trichoderma* consortium significantly increased the chitinase activity. The application of *Trichoderma* also offers many possibilities in sustainable agriculture. *Trichoderma* can not only degrade chitosan films, but also protect plants against fungal pathogens by synthesizing chitinases and *β*-1,3 glucanases with antifungal properties.

## 1. Introduction

Plastics are used in many areas such as packaging, construction materials, and hygiene products due to their chemical resistance. Unfortunately, they are not susceptible to microbial degradation, which makes them a serious environmental pollutant. For this reason, interest in biodegradable polymers has increased [1]. Chitosan obtained in the deacetylation process of chitin [2] has antimicrobial and fungicidal properties and can therefore be applied as a natural biocide in medicine, food production, and agriculture. The biocidal properties of chitosan result from its ability to degrade the cell wall and membrane, bind metal ions, and combine with DNA [3].

Chitosan-based films in particular can be used in the food packaging industry. They are modified just as often with various biocidal substances, e.g., tannic acid [4]. Phenolic substances are synthesized by plants as a variety of secondary metabolites (phytochemicals) known to be involved in defense mechanisms and, over the past few years, some of these molecules have been recognized to have health benefits, including antimicrobial properties [5]. They are capable of forming strong hydrogen bonds with polymers and thus, they are good cross-linkers for proteins and polysaccharides [6]. The presence of phenolic acids in chitosan film increases its stability for food packaging applications. Despite their use, the film polymers should be biodegradable [7,8]. Polymer degradation is a complex process involving physical, chemical, and biological factors [9]. Undoubtedly, microorganisms producing hydrolytic enzymes contribute to their biodegradation [8,10,11,12]. Fungi, for example, *Pseudozyma japonica* Y7-09 and *Trichoderma viride* 99, degraded polycaprolactone (PCL) [13,14], *Clitocybe* sp. and *Laccaria laccata* degraded polylactide (PLA) and polyethylene terephthalate (PET) [9], and *Aspergillus* degraded polyethylene (PE) [15]. Vivi et al. [16] reported that polycaprolactone (PCL) and polyvinyl chloride (PVC) were degraded by *Chaetomium globosum* ATCC 16021. A sample of each polymer was placed in a Petri dish with complete agar and a second one on the incomplete medium. A spare sample was kept as a control inside an envelope in the dark. Individual samples of polymers (PCL and PVC) were immersed in 25 mL mix suspension with a concentration of 10^6^ spores/mL for 15 s. The samples were analyzed by means of morphological and color changes, mass loss, optical microscopy (OM), and scanning electron microscopy (SEM) after 28 days of culturing. After the incubation period, visual observations of the PCL films showed many micropores and cracks, pigmentation, surface erosion and hyphal adhesion on the sample surfaces, and a mass loss of up to 75%. Also, *Cladosporium* sp., *Mucor* sp., *Penicillium caseicolun*, *Penicillium citrinium*, and *Rhizopus* sp. degraded chitosan-gellan and polylysine-gellan fibers [17]. Degradation of phenolic acid-modified chitosan film can be a problem, as native microorganisms are unable to degrade the polymers. For this reason, it is so important to introduce additional microorganisms to aid in the degradation of the chitosan film. In recent years, microbial formulations have been increasingly developed to accelerate polymers’ biodegradation. In the case of chitosan films, it is worthwhile to select chitinolytic microorganisms with multifarious enzymes which are capable of degrading chitin and its derivatives. Therefore, it is be suitable to use the fungi of the genus *Trichoderma*, which are already used in agriculture and known for their chitinolytic properties [18]. *Trichoderma* spp. belong to the Ascomycota, which are the most numerous species of fungi in almost all temperate and tropical soils. They are characterized by rapid growth, abundant sporulation, and the production of several hydrolytic enzymes [11,19].

Our previous reports in the literature indicated the possibility of the practical use of fungi like *Trichoderma* in accelerating the biodegradation of polymeric materials [9,11]. The research conducted by Znajewska et al. [14] has shown that *T. viride* accelerated the degradation of polycaprolactone in soil. Moreover, *T. viride* also showed growth on polymer materials including PE, PCL, PET, PLA, and PHB in in vitro studies [20]. Dąbrowska et al. [11] reported that the *T. viride* strain GZ1 produced hydrophobin proteins in contact with PLA and PET, producing a hydrophobin film that enabled the mycelium to adhere to the surface of the polymer.

Food packaging produced on an industrial scale from materials that are difficult to degrade represents a serious environmental problem. They take many years to decompose. An alternative to current packaging materials could be chitosan films modified with gallic, ferulic, and tannic acids. These materials can be degraded in the environment by native microorganisms. However, not all native microorganisms are able to degrade chitosan films due to their biocidal properties. In order to accelerate biodegradation, a microbiological preparation based on *Trichoderma* has been developed to support their decomposition. Fungi of the *Trichoderma* genus are known especially for the production of antifungal enzymes, which may be an additional feature when applied to the environment.

## 2. Experiments

### 2.1. Isolation and Molecular Identification of Fungal Strains

Saprophytic fungi isolated from oak wood were used in this study. The culture of the fungus stored in microbiological solid potato dextrose agar (Difco, Franklin Lakes, NJ, USA) was transferred to the same medium and incubated at 23 °C for 5 days.

Mycelia fragments were homogenized in liquid nitrogen for molecular identification. The genomic DNA from mycelia was extracted using the Plant & Fungi DNA Purification Kit (EURx, Gdansk, Poland) according to the manufacturer’s protocol. The quality and quantity of the isolated DNA were checked spectrophotometrically (Nanodrop) and electrophoretically by separation in a 1% agarose gel with ethidium bromide (BioRad, Hamburg, Germany). The isolated DNA served as a template for the PCR reaction which was run under the following conditions: The amplification reaction used a primer with the sequences (5’-CTTGGTCATTTAGAGGAAGTA-3’ and 5′-TCCTCCGCTTATTGATATGC-3′), named ITS1_f and ITS2, respectively. The genomic region including internal transcribed regions ITS2 (the region between 5.8S rRNA and 28S rRNA genes) and ITS1 (the region between 18S rRNA and 5.8 rRNA genes) was amplified [21,22]. The amplified PCR products were sequenced using the Sanger method (Genomed, Warsaw, Poland). The obtained sequences were compared with those contained in GenBank NCBI (https://blast.ncbi.nlm.nih.gov/Blast.cgi (accessed on 3 October 2022) and MycoBank Polyphasic Identifications Database (http://www.mycobank.org (accessed on 3 October 2022).

### 2.2. Chitosan Films Preparation

Phenolic acids (tannic acid, ferulic acid, and gallic acid) used for chitosan film preparation were acquired from Sigma Aldrich (Saint Louis, MO, USA). Phenolic acids were selected based on our previous research, where those compounds were used as chitosan crosslinkers [23]. Raw compounds such as chitosan and all phenolic acids were dissolved in 0.1 M acetic acid as a solvent to obtain a solution at 1% concentration. Phenolic acids were added to the chitosan (1/10 *w*/*w*%) and mixed on a magnetic stirrer. Mixtures were put on the plastic holder (40 mL per 10 × 10 cm) and left for solvent evaporation in room conditions. This resulted in obtaining thin dry films.

### 2.3. Enzymatic Characteristics of Trichoderma

The study included the determination of hydrolytic enzymes involved in the degradation of organic matter: lipase, aminopeptidase, α, and β-glucosidase. For this purpose, a liquid medium containing (g/L) sucrose 10 and yeast extract 3 was inoculated with fungal spores (10^5^/mL) and incubated for 4 days at 25 °C. The culture was then centrifuged (10 min, 10,000× *g*) and the supernatant was collected for the determination of the enzyme activity. Subsequently, 2 mL of each supernatant was added to its corresponding substrate (0.2 mL): 4-Methylumbelliferyl butyrate for lipase activity, L-Leucine-7-amido-4-methylcoumarin hydrochloride for aminopeptidase activity, 4-methylumbelliferyl α-D-glucoside, and 4-methylumbelliferyl β-D-glucoside for α and β-glucosidase activity. The final concentration of substrates in the sample was 50 µmol/mL. Incubation was carried out in a thermoblock at 40 °C for one hour. After incubation, the amount of MUF and MCA released was measured using a Hitachi F—2500 spectrophotometer [24,25,26].

Excitation/emission wavelengths were centered on 318/445 nm for methylumbelliferyl (MUF) and 345/425 for amido-4-methylcoumarinhydrochloride (MCA). The unit of activity (U) of enzymes was taken as µmol of MUF/MCA released per hour.

*Trichoderma* is known for its antifungal activity, therefore the activity of chitinase and β-1,3-glucanase (enzymes having antifungal properties) were also determined. For this purpose, a liquid medium containing (g/L) sucrose 5, yeast extract 1, and colloidal chitin 10 was inoculated with a suspension of fungal spores (10^5^/mL) and incubated at 25 °C for 4 days. After incubation, the culture was centrifuged (10 min, 10 rpm) and the activity of chitinase and β-1,3-glucanase was determined from the supernatant.

For the measurement of chitinase activity, the procedure mentioned in the above paragraph was used, taking methylumbelliferyl-N-acetyl-β-D-glucosaminide as the substrate [25]. The β-1,3-glucanase activity was determined according to Wu et al. [27] by the spectrophotometric method. In brief, 500 µL of supernatant was mixed with 500 µL of laminarin 0.5% (*v*/*w*) in 100 mM sodium acetate buffer (pH 5.5). The reaction was allowed to proceed at 50 °C for 60 min after which it was terminated by heating for 5 min at 100 °C. Thereafter, 2 mL of dinitrosalicylate (DNS) 1% (*v*/*w*) was added to the reaction and the mixture was boiled for 10 min. After cooling the samples, the sugar concentration was measured with a spectrophotometer at 540 nm. The β-1,3-glucanase of activity was expressed as μmoL of glucose per hour.

### 2.4. Biological Activity of Trichoderma in the Presence of Chitosan Films

The biological activity of *Trichoderma* in the presence of the test films was determined using the respirometric method with the OxiTop Control system (WTW, Wroclaw, Poland) according to the modified method of Swiontek Brzezinska et al. [28]. Into the OxiTop measuring dishes, 100 mL of sterile medium containing (g/L) sucrose 5, yeast extract 0.5, and fragments (2 cm × 2 cm) of the test films (total mass 1 g) were placed and inoculated with 0.1 mL of *Trichoderma* suspension (spores concentration 10^5^/mL). The quivers with CO_2_ absorbent (0.4 g NaOH) were placed in the measuring dishes and the measuring heads were screwed on. Incubation was carried out at 26 °C for 14 days. The control sample contained a polymer-free medium inoculated with the same amount of fungal suspension. The biological activity was expressed as mg of O_2_/L of culture.

### 2.5. Fourier Transform Infrared Spectroscopy—Attenuated Total Reflectance (FTIR–ATR)

FTIR–ATR analysis was made for each kind of chitosan film modified by phenolic acids. Films before and after degradation were placed under the diamond ATR crystal and scanned with 4 cm^−1^ resolution in the range of 4000–600 cm^−1^ using Nicolet iS10 spectrometer (Thermo Fisher Scientific Inc., Waltham, MA, USA). Each registered spectrum was obtained by averaging 64 scans. Spectra were recorded and graphs were prepared in the Excel program. Only the CTS/GA and CTS/TA films were selected for testing as the other films were broken and were not suitable for further analysis.

### 2.6. Mechanical Properties

The mechanical testing was carried out using a testing machine (Shimadzu EZ-Test EZ-SX, Kyoto, Japan). Samples were inserted between two handles and stretched (5 mm/min). Young’s Modulus was calculated from the slope of the stress–strain curve in the linear region with the use of the Trapezium X Texture program. The thickness of the samples was measured using a thickness gauge (Sylvac, Yverdon-les-Bains, Switzerland). Each measurement was carried out in ten repetitions [6].

### 2.7. Scanning Electron Microscopy (SEM)

The morphology of the samples was studied using Scanning Electron Microscope (SEM) (LEO Electron Microscopy, Thornwood, NY, USA). The films were covered with gold before the observation. Scanning electron microscope images were made with a magnification of 1000×.

### 2.8. Biodegradation of Chitosan Films in the Compost after Application of Trichoderma Strains

The biodegradation of the chitosan films in the compost after the application of *Trichoderma* strains was determined using the OxiTop system [29] according to the modified procedure described previously. For this purpose, 100 g of compost was placed in the jar of the OxiTop system, mixed with chitosan film fragments (total mass, 1 g, 2 cm × 2 cm), and inoculated with a 1 mL fungal culture (10^9^ CFU/mL). Incubation was carried out at 26 °C for 21 days.

The study was carried out in variants: variant 1—biodegradation of chitosan films by native compost microorganisms; variant 2—biodegradation of chitosan film after application of *T. atroviride* TN1; variant 3—biodegradation of chitosan film after application *T. citrinoviride* TN3; variant 4—biodegradation of chitosan film after application of consortium composed of TN1 and TN3. The control sample contained compost without chitosan films and without supplementation with a fungal culture. The susceptibility to biodegradation of chitosan films was determined based on oxygen consumption and expressed as mgO_2_/kg compost. The physicochemical parameters of the compost are shown in Table 1. The pH of the compost was measured by the potentiometric method using a glass electrode in 1 M KCl. The Tiurin procedure was used to assess total organic carbon (TOC), while total nitrogen (TN) was determined by the Kjeldahl method [30]. Soil humidity was measured with a tensionometer T1-q-30. P_2_O_5_, K_2_O, Mg, N-NO_3_, and NNH_4_ were determined according to the following standards: PN-ISO 10390:1997, PN-R-04022:1996, PN-R-04022:1996/Az1:2002, PN-R-04020:1994/Az1:2004, and PN-R-04028:1997 [31,32,33,34,35]. Biological oxygen consumption (BOD) at 20 °C for 5 days was measured using the OxiTop measurement system [36]. Hydrolase activity was checked using fluorescein diacetate (Sigma-Aldrich) by following Adam & Duncan’s [37] method, with a small modification. Microbial abundance was tested using the Koch plate method. Compost samples, after a series of tenfold dilutions, were transferred to nutrient agar (for bacteria), potato glucose agar (for molds), and actinomycetes agar (for actinomycetes).

### 2.9. Effect of Application of Trichoderma on Hydrolytic Enzymes Activity in Compost

Compost samples without chitosan films (control), compost samples containing chitosan films without fungal supplementation, and compost samples with chitosan films after bioaugmentation with a *Trichoderma* preparation were used for the study. To determine the enzymatic activity, 10 g of compost was ten-fold diluted using saline (0.9% NaCl) and shaken for 20 min. The activities of aminopeptidase, lipase, chitinase, α, and β-glucosidase were determined in the diluted samples as described previously (point 2.3).

### 2.10. Statistical Analysis

The statistical analyses of studies in 3.2, 3.3, and 3.8 point were conducted using Past v. 3.08 [38], with three replicates for each type of data. For all datasets, we calculated summary statistics (mean and standard deviation), performed a normality test (Shapiro–Wilk test), and assessed homogeneity of variance (Levene’s test). One-way ANOVA was employed to examine differences in enzyme activity and BOD between sample groups with different *Trichoderma* strains. Tukey’s pairwise test was applied for post hoc comparisons.

The statistical analysis of the rest of the study was performed using commercial software (SigmaPlot 14.0, Systat Software, San Jose, CA, USA). The Shapiro–Wilk test was used to assess the normal distribution of the data. All the results were calculated as means ± standard deviations (SD) and statistically analyzed using one-way analysis of variance (one-way ANOVA). Multiple comparisons versus the control group between means were performed using the Bonferroni *t*-test with the statistical significance set at *p* < 0.05.

## 3. Results and Discussion

### 3.1. Molecular Identification of Trichoderma

Conventional methods for the identification of *Trichoderma* spp. using cultural and morphological methods might provide erroneous results [39]. A better approach is to use molecular phylogenetic characteristics in combination with conventional methods [40]. One of the molecular methods used to identify fungi is rDNA sequencing [41,42,43] identification which has made it possible to identify fungi belonging to the species of *Trichoderma atroviride* and *Trichoderma citrinoviride*. The resulting sequences designated TN1–3 have been deposited with GeneBank NCBI under accession numbers (TN1—ID OP586643.1; TN2—ID OP586653.1 and TN3—ID: OP586652.1). A comparison of the TN1 sequence with the sequences available from MycoBase showed homologies to *T. atroviride* and *T. citrinoviride*. The TN1 sequence showed homology to *T. atroviride* (NBRC 101776) with 99.82% sequence similarity, TN2 was 99.83% homologous to *T. atroviride* (FMR12650), and TN3 sequence showed 100% similarity to *T. citrinoviride* (DAOM 172792). Identification of fungi is often problematic and it is not always possible to identify them by species [43]. There are various molecular characterization techniques to confirm *Trichoderma* isolates, such as amplification and sequence analysis of the internal transcribed spacer gene ITS 1 and 2 and the translation elongation factor 1-alpha (tef1) encoding gene and BLAST interface in TrichOKEY and TrichoBLAST (https://trichokey.com/index.php/trichoblast (accessed on 3 October 2022) [39]. Dendrogram prepared using nucleotide sequences of the ITS region of the 18S rRNA gene and using Clustal W showed in Appendix A.

### 3.2. Enzymatic Activity of Trichoderma Strains

The results of enzymatic activities of three different *Trichoderma* cultures designated as TN1, TN2, and TN3 are presented in Table 2.

Strain *T. atroviride* TN1 and *T. citrinoviride* TN3 appeared to be the most enzymatically active. All the *Trichoderma* strains were characterized by the highest lipase and aminopeptidase activity. These enzymes are important in the degradation of organic matter, which can be deposited in significant quantities in the environment. All the tested isolates produced chitinases and β-1,3-glucanases, which are known for their fungicidal properties. By producing fungicidal chitinases, *Trichoderma* strains are used for plant protection.

Also, some secondary metabolites and hydrolytic enzymes excreted by *Trichoderma* have been shown to be involved in inhibiting the growth of pathogenic microorganisms and stimulating plant growth [44]. Cherkupally et al. [45] studied the *Trichoderma* genus for activity on amylases, cellulases, peptidases, pectinases, and chitinases. They reported that *T. harzianum*, *T. pseudokoningii*, *T. atroviride*, *T. viride*, *T. virens*, and *T. koningii* had high chitinolytic activity. Similarly, Xia et al. [46] found that *T. viride* showed high chitinase activity. Win et al. [47] isolated *Trichoderma asperellum*, which presented antifungal activity against four species of phytopathogenic fungi from the *Fusarium* genus. In addition, *Trichoderma asperellum* has been shown to secrete mycolytic enzymes (chitinase and β-1,3-glucanase). The number of chitinase genes showed great variation in fungal genomes, ranging from a single gene in *Schizosaccharomyces pombe* to 36 genes in *T. virens* [48].

*Trichoderma* have the ability to detoxify pesticides and herbicides [49]. This is certainly due to the processing of substances by extracellular and intracellular enzymes. According to Abdenaceur et al. [50], *Trichoderma* isolates obtained in the ecosystem niche of olives showed high potential for plant growth promoting biomolecule production. According to Wang et al. [51], *Trichoderma lentiforme* ACCC30425 showed the highest lipase production capacity at high pH. According to Qian et al. [52], the filamentous fungus *Trichoderma reesei* is one of the best-known cellulolytic organisms and the significant producer of cellulases for industrial applications. In addition, aspartic protease P6281 secreted by *Trichoderma harzianum* is important in mycoparasitism on phytopathogenic fungi [53].

### 3.3. Respirometric Activity of Trichoderma in the Presence of Modified Chitosan Films

The tested *Trichoderma* strains degraded chitosan films at different levels (Table 3).

*T. atroviride* TN1 and *T. citrinoviride* TN3 metabolized the tested films best. At the same time, a statistically significant influence of phenolic acids incorporated into chitosan on the biological activity of *Trichoderma* was observed. CTS/FA films were best used by *Trichoderma* (BOD 130–257 mg O_2_/L after 8 days). In turn, CTS/GA films were less prone to biodegradation (BOD 98–117 mg O_2_/L after 8 days).

Chitosan has many intrinsic characteristics including biocompatibility and biodegradability [54]. Furthermore, it shows antimicrobial, antitumor, and antioxidant properties [55]. Inhibition of fungal and bacterial growth depends on the molecular weight of chitosan and its functional groups. Smaller oligomeric chitosan molecules can easily penetrate the cell membrane and inhibit RNA transcription to prevent cell growth [56].

Our previous research [4] showed that chitosan film modified with tannic acid was difficult to degrade by microorganisms. However, there is no doubt that there might be some microorganisms in the environment capable of utilizing chitosan as a source of carbon. Ohkawa et al. [17] studied the degradation of chitosan–gellan (CGF) by seven species of soil filamentous fungi. Biodegradation processes were observed under the microscope and it was revealed that only *P. caseicolum* on the CGF grew along with the breakdown of the fiber matrices. In the biochemical oxygen demand (BOD) test, the degradation of CGF by *A. oryzae* was the highest at 59% among the seven species of tested fungi. *Penicillium caseicolum* and *P. citrinum* also degraded the CGF at the level of 41% and 10%, respectively. Babaee et al. [57] examined the biodegradation of chitosan nanoparticles (CNPs) films using *T. versicolor*. According to these authors, the chitosan films underwent a fast degradation process during storage within the first 30 days and complete degradation after 50–60 days.

### 3.4. Fourier Transform Infrared Spectroscopy—Attenuated Total Reflectance (FTIR–ATR)

The degradation of chitosan films modified by phenolic acids was observed in all studied conditions (Figure 1A), however, it was not controlled by the type of phenolic acid added to chitosan. The degradation peak in the range of 3600–2700 cm^−1^ was observed from amine and hydroxyl groups. A peak was not observed after the degradation. Similarly, a peak at 1025 cm^−1^ from COH and CH_2_OH groups was observed before degradation and was not noticed after the treatment with fungi. It suggested that the degradation process broke C-N and C-O bonds. Peaks at 2055 and 1835 cm^−1^ were present after degradation. It proved that the number of C=C and C=O groups had increased. The characteristic peaks of chitosan films modified by tannic acid are shown in Figure 1B. Similar observations were also found for CTS/GA acid films (Figure 1C).

Through previous studies, it has been shown that chitosan can be modified by phenolic acids addition. The FTIR spectra of chitosan-based materials with ferulic acid, gallic acid, and tannic acid were published before by Kaczmarek-Szczepańska et al. [21]. Significant changes in the localization and the presence of characteristic groups on the spectra were not observed. It was assumed that covalent bonds between chitosan and phenolic acids are not formed, and the cross-linking process is based on the hydrogen bond formation between hydrophilic groups.

Stoleru et al. [58] tested the biodegradation of poly(lactic acid)/chitosan by *Phanerochaete chrysosporium*, a white rot fungus. The degradation was followed by infrared spectroscopy (FTIR). Notable structural changes resulted from the fungal biodegradation of PLA/chitosan-based samples which showed the onset of hydrolytic or fungal degradation. Biodegradation by the fungi increased the crystallinity of the samples, which means that degradation occurs primarily in amorphous regions. It should be noted that chitosan in the initial days of incubation may act as an antimicrobial agent and the fungi tested may be less active in degrading chitosan films. Stoleru et al. [58] reported that changes in structure are less pronounced for polylactide/chitosan composite than for uncoated polylactide which undergoes important changes after a short incubation period. Therefore, after degradation in the presence of P. chrysosporium, due to hydrolytic cleavage of polylactide backbone and chitosan deacetylation, both components of the layered composites exhibited structural changes. Our previous research [4] showed that bacteria isolated from chitosan films modified by tannic acid accelerated the degradation of the materials. Before the degradation, films based on chitosan and tannic acid in a ratio of 80/20 had a big peak at 1289 cm^−1^ from the C-O-C group, which corresponded to the backbone of the chitosan polymeric chain. It decreased after biodegradation. This may be due to the fact that the bond is broken by the action of microorganisms.

### 3.5. Mechanical Properties

All tested samples after the degradation process showed greater thickness than before (Table 4).

As a result of degradation, the film shrinks, thereby allowing the increase in thickness to be observed. Mechanical parameters such as Young Modulus, maximum tensile strength, and elongation at break were determined (Figure 2).

It was noticed that the highest value of Young Modulus and maximum tensile strength were noticed for the material composed of chitosan and ferulic acid (Figure 2). Mathew & Abraham [59] showed that the incorporation of ferulic acid to starch/chitosan films enhances the tensile strength. In terms of gallic acid incorporated into chitosan/gelatin, the chitosan-based films modified by gallic acid showed higher Emod and σmax values than the films without phenolic acids. Similar results were obtained by Rui et al. [60] where gallic acid addition resulted in the increase in mechanical properties of the fabricated films. No significant differences in the elongation at break values were observed between the studied samples. Qin et al. [61] observed that the addition of tannic acid to chitosan results in an increase in tensile strength as well as the Young Modulus. 0Also, Sun et al. [62] documented that the presence of gallic acid results in a significant increase in the tensile strength of the studied chitosan/gallic acid films. No statistically significant differences in the elongation at break values were observed between the studied samples. Similar results were obtained by Rivero et al. [63] where the elongation at break did not significantly change after the addition of phenolic acids.

### 3.6. Scanning Electron Microscopy (SEM)

The images of CTS/TA and CTS/GA before degradation showed the smooth surface of films without any scratches or structures (Figure 3).

It can be observed that after degradation in the compost, the soil grains are present on the surface of both types of films. For images after biodegradation in the compost after application, the *Trichoderma* application allowed us to observe the presence of fungi-like structures that adhere to the materials’ surface. It suggests that the biodegradation process includes the adhesion of fungi to the CTS/TA and CTS/GA surface, both TN1 and TN3 types. Similar structures were observed by Poorna Chandrika et al. [64] on SEM images of chitosan-PEG blended films using *Trichoderma* and by Cheng et al. [65] on carvacrol/sodium alginate films.

### 3.7. Biodegradation of Chitosan Films in the Compost after Application of Trichoderma 

The biodegradation of the chitosan film modified by phenolic acids after application of *Trichoderma* is shown in Figure 4.

Overall, the oxygen consumption by microorganisms in the presence of chitosan films modified by phenolic acids was significantly dependent on the type of phenolic acids used, the fungal culture, and the interaction of the two factors (Table 5). 

The application of *T. atroviride* TN1 and *T. citrinoviride* TN3 similarly influenced the degradation of the CTS/TA film. The respiratory activity after 21 days of incubation was 492 mgO_2_/kg and 468 mgO_2_/kg. However, the highest oxygen consumption in the presence of CTS/TA film was found after the application of the consortium of these strains (861 mg O_2_/kg after 21 days of incubation). Similarly, chitosan with gallic acid (CTS/GA) was least degraded by the native microorganisms of the compost (Figure 4B). The application of *T. atroviride* TN1 and *T. citrinoviride* TN3 similarly influenced the degradation of the CTS/GA film. The respiratory activity was 553 mgO_2_/kg and 566 mgO_2_/kg, respectively, after 21 days of incubation. However, the highest oxygen consumption in the presence of CTS/GA was found after the application of the consortium of these strains (849 mgO_2_/kg after 21 days of incubation). Similar relationships were observed after the introduction of chitosan with ferulic acid (CTS/FA) into the compost after the application of the *Trichoderma* consortium (Figure 4C). The films before and after degradation in Oxi-Top are shown in Figure 5.

Composting is a significant method of waste management, including polymer films. Altun et al. [66] used controlled composting reactors to study the biodegradation of chitosan films. Microbial diversity was also determined by PCR-denaturing gradient gel electrophoresis (DGGE). The dominant taxonomic groups of microorganisms in composting reactors at the phylum level were Ascomycota and Proteobacteria. Carbon dioxide emissions data showed a high degradation rate of chitosan in reactors inoculated with microorganisms, which were capable of producing chitosanase. The control reactor contained only chitosan films (no microbial inoculum). In comparison, the group I microorganisms (*Bacillus circulans*, *Bacillus licheniformis*, *Streptomyces roseolus*, *Streptomyces zaomyceticus*, *Penicillium islandicum*, and *Penicillium chrysogenum*) increased the biodegradation rate. On the other hand, the group II inoculant (*Mycoplana dimorpha*, *Fennellia flavipes*, and *Chaetomium globosum*) reduced the efficiency of chitosan biodegradation. In the biodegradation of chitosan, the appropriate selection of microbial strains for inoculation is important, especially strains producing chitinases of different subclasses. Kaczmarek-Szczepańska et al. [4] reported that the bacteria isolated from the soil degraded the chitosan films modified by tannic acid (50CTS/50TA) better than the bacteria isolated from the compost. Their study also showed that consortia of bacteria (*Acinetobacter calcoaceticus* GTA2, *Bacillus mycoides* GTA7, *Pseudomonas arsenicoxydans* GTA8, and Pseudomonas laurylsulfativorans GTA9) added to soil had a positive effect on the biodegradation of the 50CTS/50TA. Similarly, consortia of bacteria (*Pseudomonas hunanensis* KTA1 and *Pseudomonas psychrophila* KTA3) added to the compost accelerated the degradation of chitosan films modified by tannic acid (50CTS/50TA).

Our study showed that *T. atroviride* metabolized chitosan-based materials, but the process was relatively slow. The chitosan films used for this study were modified with phenolic acids, which had antimicrobial properties. Therefore, it can be speculated that the degradation of such materials might start with the adaptation of microorganisms to the polymers and the production of enzymes such as chitosanase. Znajewska et al. [14] reported that *T. viride* DAR5, *T. viride* 3333, and *T. viride* 154 accelerated polycaprolactone (PCL) degradation in the soil. The greatest degree of polymer degradation expressed in terms of mass loss was shown by *T. viride* 3333 (complete disintegration of the PCL film). The other two strains degraded PCL to a lesser extent (PCL mass loss of 27.85% and 11.58%, respectively). These results were obtained after 6 months of PCL storage in the soil. On the other hand, the studies by Dąbrowska et al. [11] showed that *T. viride* GZ1 is capable of producing hydrolytic enzymes degraded polylactide and polyethylene terephthalate, producing hydrophobin in the first stage of biodegradation.

The introduction of engineered consortia of microorganisms into the environment often has a positive effect not only on the biodegradation processes of organic matter but also improves soil quality and favorably influences plant development. For example, the addition of suitable microorganism-based boosters compensates for stress and reduced plant growth caused by weeds, drought, heavy metals, salt, and other adverse environmental conditions [67].

### 3.8. Effect of Application of Trichoderma on Hydrolytic Enzymes Activity in Compost

Extracellular enzymes secreted by microorganisms into the compost significantly affect the degradation of organic matter. The introduction of the *Trichoderma* preparation into the compost further enriched the compost with enzymes capable of degrading macromolecular compounds (Table 6).

Lipase and aminopeptidase were found to be the most active. The greatest changes in enzyme activity were found for chitinases. These enzymes showed the highest activities after the introduction into the compost chitosan films with ferulic acid. The application of the *Trichoderma* consortium significantly increased the activity of chitinases in the compost. In contrast, the activity of glucosidases in the compost was the lowest even after the introduction of chitosan films and *Trichoderma*.

The introduction of materials with antimicrobial properties into the environment often affects the production of extracellular enzymes by microorganisms. And our results also showed that for most enzymes, inhibition of activity was observed after the introduction of chitosan films with ferulic, tannic, and gallic acids. Further, the addition of the *Trichoderma* consortium to the compost improved enzyme synthesis, especially chitinase, lipase, and aminopeptidase. Klose et al. [68] fumigated toxic compounds including methyl bromide, propargyl bromide, chloropicrin, and sandy soil, and then checked the activity of acid phosphatase, arylsulfatase, β-glucosidase, and dehydrogenase. Their tests showed that arylsulfatase (62%), dehydrogenase (35%), acid phosphatase (22%), and β-glucosidase (6%) activities decreased within 90 days after fumigation. Su et al. [69] studied enzyme activity in arsenic-contaminated soils. After inoculation of *Trichoderma asperellum* SM-12F1 into contaminated soil, they observed β-glucosidase (155%), chitinase (211%), and phosphatase (108%) activities in SM soils, while significantly decreased β-glucosidase (81%), phosphatase (54%), aminopeptidase (60%), and catalase (67%) activities in soils. Such a result demonstrates the positive effects of bioaugmentation, which is increasingly practiced. Inoculation of *Trichoderma* into the environment can improve enzymatic activity in the rhizosphere of plants. Yedidia et al. [70] reported that cucumber roots inoculated with *Trichoderma harzianum* T-203 showed higher chitinase, β-1,3-glucanase, cellulase, and peroxidase activities up to 72 h after inoculation compared to roots without *Trichoderma.* Soil enzyme activity is sensitive to environmental changes and has been proposed as an indicator of soil quality due to its relationship with soil biology [71] and rapid responses to changes in management and soil environment [72,73]. Therefore, the introduction of biocidal materials into the environment is associated with the inhibition of enzyme activity. Inoculation with microorganisms with high enzymatic activity can improve soil quality as well as soil enzyme activity.

## 4. Conclusions

Chitosan is known primarily for its antifungal properties. Chitosan-based liquid preparations have been developed for plant protection. Chitosan films are also used in medicine as a dressing material. The introduction of biocidal phenolic acids into chitosan film makes it possible to use it as food packaging to protect against the growth of both bacteria and fungi. The potential of modified chitosan films may be seen in the protection of fruits and vegetables. Such foil can protect them against the development of fungi. However, packaging materials should be degradable in the environment. Studies have shown that chitosan modified with phenolic acid is less readily degraded by native microorganisms. Only after the application of *Trichoderma* does the degradation of chitosan films occur more effectively. At the same time, studies have shown a beneficial effect of the use of *Trichoderma* on the enzymatic activity of compost. Chitinases, the activity of which increased after the introduction of the preparation, can control the development of phytopathogens.

## Figures and Tables

**Figure 1 foods-12-03669-f001:**
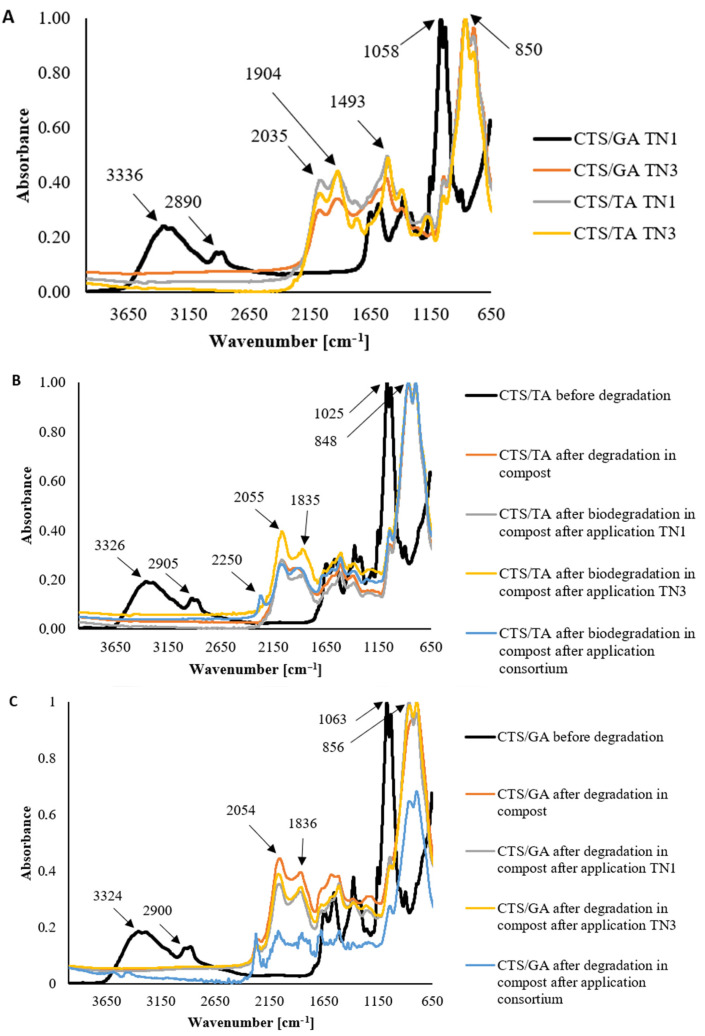
The FTIR–ATR spectra of (**A**) chitosan film modified by gallic acid (CTS/GA) and tannic acid (CTS/TA) after degradation of *Trichoderma* strains (TN1 and TN3); chitosan film modified by tannic acid (**B**) and gallic acid (**C**) before and after degradation in compost after application *Trichoderma* strains (TN1 and TN3).

**Figure 2 foods-12-03669-f002:**
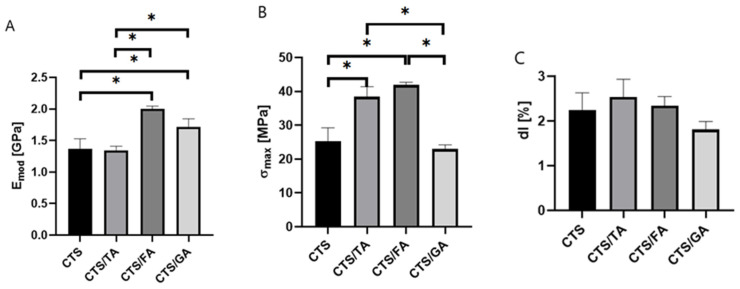
The Young Modulus (**A**), maximum tensile strength (**B**), and elongation at break (**C**) of films based on chitosan (CTS) with tannic acid (TA), ferulic acid (FA), and gallic acid (GA) (* significantly different *p* < 0.05).

**Figure 3 foods-12-03669-f003:**
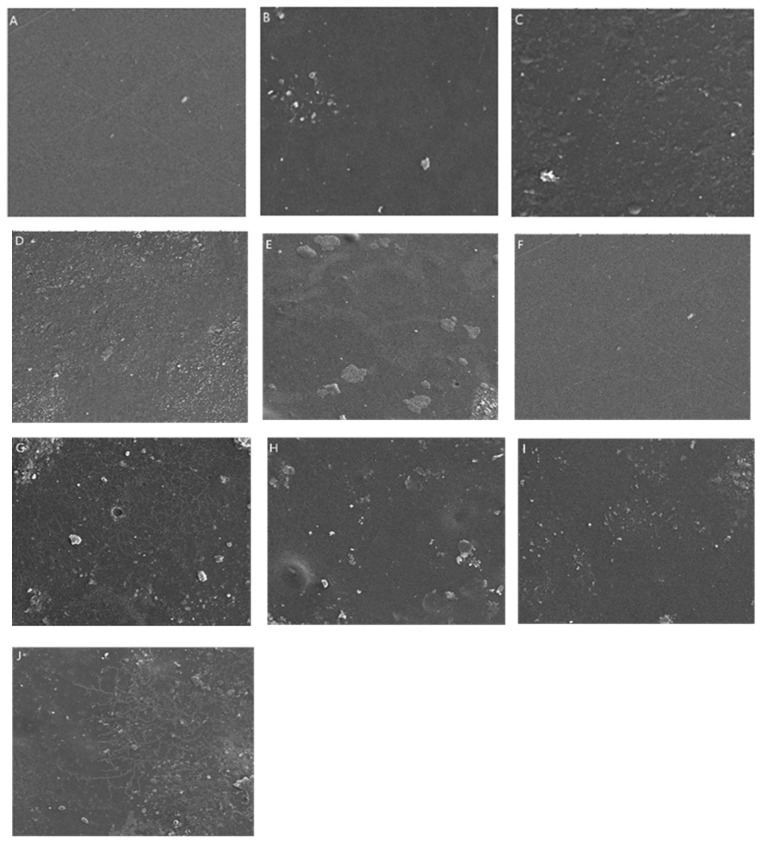
The SEM images (1000×) of (**A**) CTS/TA before degradation; (**B**) CTS/TA after degradation in compost; (**C**) CTS/TA after biodegradation in compost after application of TN1; (**D**) CTS/TA after biodegradation in compost after application of TN3; (**E**) CTS/TA after biodegradation in compost after application of consortium; (**F**) CTS/GA before degradation; (**G**) CTS/GA after degradation in compost; (**H**) CTS/GA after biodegradation in compost after application of TN1; (**I**) CTS/GA after biodegradation in compost after application of TN3; (**J**) CTS/GA after biodegradation in compost after application of consortium.

**Figure 4 foods-12-03669-f004:**
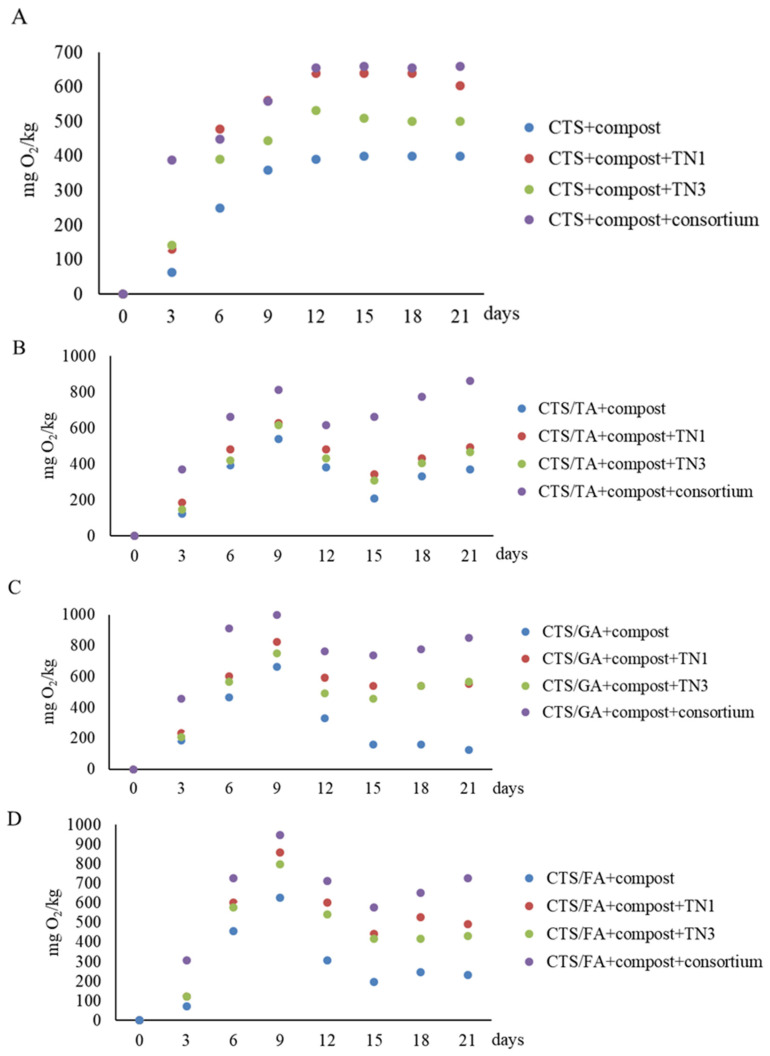
The biodegradation of chitosan film modified by phenolic acids after application of *Trichoderma* strains (TN1 and TN3): chitosan (**A**), chitosan modified tannic acid (**B**), chitosan modified gallic acid (**C**), and chitosan modified ferulic acid (**D**).

**Figure 5 foods-12-03669-f005:**
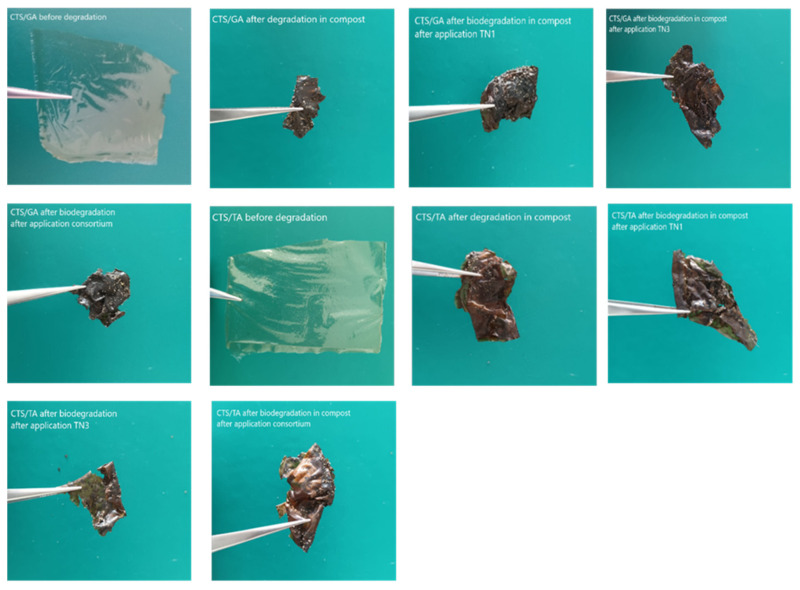
The films before and after degradation in OxiTop.

**Table 1 foods-12-03669-t001:** Physicochemical properties of the compost. Values are expressed as mean ± SD (*n* = 3).

Parameters	Values
pH	7.5
humidity	200 mbar/hPa
TOC (g/kg)	351
TN (g/kg)	27
N-NH_4_ (mg/kg)	2.72
N-NO_3_ (mg/kg)	29.0
P_2_O_5_ (mg/100 g)	26.2
Mg (mg/100 g)	7.1
K_2_O (mg/100 g)	27.5
BOD_5_ (mgO_2_/kg)	80
Hydrolase activity (mg/kg)	14
Heterotrophic bacteria numbers (CFU/g)	45 × 10^5^
Actinomycete numbers (CFU/g)	39 × 10^4^
Mold numbers (CFU/g)	23 × 10^3^

**Table 2 foods-12-03669-t002:** Enzymatic activity of *Trichoderma* strains. Values are expressed as mean ± SD (*n* = 3). Different letters in the same line indicate statistically significant differences with *p* < 0.05).

Enzymes Activity (U/h)	*Trichoderma* *artoviride* TN1	*Trichoderma* *artoviride* TN2	*Trichoderma* *citrinoviride* TN3
chitinase	0.82 ± 0.03 ^a^	0.4 ± 0.03 ^c^	0.70 ± 0.04 ^b^
β—1,3—glucanase	1.63 ± 0.02 ^a^	0.7± 0.02 ^b^	1.9 ± 0.01 ^a^
lipase	2.23 ± 0.01 ^a^	0.51 ± 0.20 ^c^	1.99 ± 0.06 ^b^
α-glucosidase	0.08 ± 0.01 ^a^	0.06 ± 0.01 ^b^	0.08 ± 0.04 ^a^
β-glucosidse	0.05 ± 0.01 ^b^	0.02 ± 0.01 ^c^	0.09 ± 0.02 ^a^
aminopeptidase	1.7 ± 0.02 ^b^	1.9 ± 0.01 ^a^	1.8 ± 0.01 ^a^

**Table 3 foods-12-03669-t003:** Biological oxygen demand of *Trichoderma* strains in the presence of chitosan films modified by phenolic acids.

Materials	*Trichoderma* *artoviride* TN1	*Trichoderma* *artoviride* TN2	*Trichoderma* *citrinoviride* TN3
CTS	* 137 ± 2.2 ^a^	89 ± 3.3 ^a^	110 ± 2.5 ^a^
CTS/TA	210 ± 3.1 ^b^	120 ± 3.4 ^b^	178 ± 3.2 ^b^
CTS/GA	104 ± 4.0 ^a^	98 ± 1.3 ^a^	117 ± 3.3 ^a^
CTS/FA	257 ± 4.2 ^c^	130 ± 3.2 ^b^	220 ± 4.1 ^c^

* Biological activity was expressed in mg O_2_/L after 8 days of incubation. Values are expressed as mean ± SD (*n* = 3). Different letters in the columns indicate statistically significant differences with *p* < 0.05).

**Table 4 foods-12-03669-t004:** The thickness of films based on chitosan (CTS) with tannic acid (TA), ferulic acid (FA), and gallic acid (GA) before and after degradation (significantly different *p* < 0.05 from control: ^a^ vs. CTS, ^b^ vs. CTS/TA, ^c^ vs. CTS/GA).

Sample	Thickness [mm]	Sample	Thickness [mm]
CTS	0.090 ± 0.009	CTS/TA after degradation in compost	1.073 ± 0.011 ^a,b^
CTS/TA	0.131 ± 0.008 ^a^	CTS/TA after degradation in compost after application TN1	1.038 ± 0.017 ^a,b^
CTS/FA	0.100 ± 0.010	CTS/TA after degradation in compost after application TN3	1.098 ± 0.010 ^a,b^
CTS/GA	0.080 ± 0.006	CTS/TA after degradation in compost after application consortium	1.671 ± 0.013 ^a,b^
		CTS/GA after degradation in compost	1.011 ± 0.008 ^c^
		CTS/GA after degradation in compost after application TN1	1.045 ± 0.014 ^c^
		CTS/GA after degradation in compost after application TN3	1.024 ± 0.009 ^c^
		CTS/GA after degradation in compost after application consortium	1.781 ± 0.018 ^c^

**Table 5 foods-12-03669-t005:** Two-way ANOVA results on the influence of phenolic compounds, fungal strain/consortium (experiment variant), and their interaction on biological oxygen demand (BOD 21) in compost.

	Sum of Squares	df	Mean Square	F	*p*-Value
Phenolic substance	4.48 × 10^4^	3	1.49 × 10^4^	1.25 × 10^3^	3.32 × 10^−33^
Variant	1.46 × 10^6^	3	4.88 × 10^5^	4.08 × 10^4^	2.24 × 10^−57^
Interaction	2.46 × 10^5^	9	2.74 × 10^4^	2.29 × 10^3^	2.55 × 10^−42^
Within	3.83 × 10^2^	32	1.20 × 10^1^		
Total	1.76 × 10^6^	47			

**Table 6 foods-12-03669-t006:** Effect of chitosan film modified by phenolic acids on changes in soil hydrolytic enzyme activity after *Trichoderma* strains bioaugmentation.

Variants	Enzyme Activity (U/h)				
	Chit.	Lip.	α-Gluc.	β-Gluc	Amino.
Compost	4.4 ±0.14 ^a^	19.4 ± 0.1 ^a^	4.2 ±0.13 ^a^	4.2 ± 0.03 ^a^	8.2 ± 0.05 ^a^
CTS/TA + compost	3.1 ± 0.03 ^b^	15.6 ± 0.01 ^b^	4.5 ± 0.01 ^a^	4.5 ± 0.01 ^a^	6.8 ± 0.05 ^b^
CTS/TA + compost + TN1	5.8 ± 0.07 ^c^	16.8 ± 0.10 ^b^	4.5 ± 0.01 ^a^	4.3 ± 0.03 ^a^	9.2 ± 0.06 ^c^
CTS/TA + compost + TN3	5.3 ± 0.06 ^c^	17.9 ± 0.07 ^b^	4.4 ± 0.01 ^a^	3.9 ± 0.03 ^a^	9.6 ± 0.02 ^c^
CTS + compost + TN1 + TN3	5.8 ± 0.21 ^c^	34.2 ± 0.31 ^c^	4.6 ± 0.41 ^a^	4.2 ± 0.02 ^a^	9.4 ± 0.06 ^c^
CTS/GA + compost	4.0 ± 0.18 ^a^	7.0 ± 0.05 ^b^	3.6 ± 0.01 ^a^	4.1 ± 0.01 ^a^	1.3 ± 0,03 ^b^
CTS/GA + compost + TN1	5.5 ± 0.01 ^a^	13.2 ± 0.12 ^c^	3.7 ± 0.01 ^a^	4.2 ± 0.02 ^a^	5.8 ± 0.01 ^c^
CTS/GA + compost + TN3	6.6 ± 0.03 ^b^	13.4 ± 0.01 ^c^	3.0 ± 0.01 ^a^	4.3 ± 0.02 ^a^	6.2 ± 0.01 ^c^
CTS/GA + compost + TN1 + TN3	11.2 ± 0.12 ^c^	13.6 ± 0.01 ^c^	3.0 ± 0.01 ^a^	3.9 ± 0.01 ^a^	5.4 ± 0.05 ^c^
CTS/FA + compost	3.9 ± 0.28 ^a^	19.9 ± 0.08 ^a^	4.7 ± 0.01 ^a^	4.2 ± 0.08 ^a^	3.7 ± 0.02 ^b^
CTS/FA + compost + TN1	25 ± 0.11 ^b^	14.0 ± 0.04 ^b^	4.8 ± 0.01 ^b^	4.4 ± 0.01 ^a^	7.4 ± 0.03 ^c^
CTS/FA + compost + TN3	28.3 ± 0.28 ^b^	23.3 ± 0.01 ^b^	4.6 ± 0.03 ^b^	4.9 ± 0.01 ^a^	6.5 ± 0.03 ^c^
CTS/FA + compost + N1 + TN3	38.5 ± 0.57 ^c^	25.1 ± 0.01 ^b^	4.3 ± 0.02 ^b^	4.6 ± 0.01 ^a^	6.1 ± 0.06 ^c^

Values are expressed as mean ± SD (*n* = 3). Chit—chitinase, Lip—lipase, α-gluc—α-glucosidase, β-gluc—β-glucosidase, and Amino—aminopeptodase. Different letters in the column indicate statistically significant differences at *p* < 0.05 according to the HDS Tukey’s test.

## Data Availability

The data used to support the findings of this study can be made available by the corresponding author upon request.

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
