# Peer review of "Application Potential of Trichoderma in the Degradation of Phenolic Acid-Modified Chitosan"

_foods, 2023, doi:10.3390/foods12193669_

Round 1
Reviewer 1 Report
In this manuscript the authors investigated the potential use of fungi of the genus Trichoderma for the degradation of phenolic acid-modified chitosan in compost. The results showed that the application of Trichoderma had a beneficial effect on the enzymatic activity in the compost.
The environmental pollution caused by plastics has been an important scientific and engineering issue. Thus, this is an important topic and the purpose of the manuscript is valuable. The paper is well written and clear, and need a minor revision.
The minor issues should be addressed:
1. Line 334, “3.4. Fourier transform infrared spectroscopy – attenuated total reflectance.” I find it difficult to locate the peak positions described by the author in the figure. I suggest that the author annotate the figure with the functional groups corresponding to the peaks or indicate the peak values explicitly.
2. 3600-2700 /cm. The notation of this unit is not standard and needs to be corrected. “3600-2700 cm-1”
3. Line 424. “Figure 3. The SEM images.” The figure caption describes Figure 3a, 3b, and 3c, but Figure 3a, 3b, and 3c are not included in the figure.
4. Line 102. “0.1M” should be “0.1 M”.
5. Line 162-163. “with 4 cm−1 resolution, in the range 4000-600 cm-1”. The “-1” in the unit should be superscript.
6. Line 329. “10 %” should be “10%”.
Author Response
Dear All,
on behalf of myself and co-authors, I am enclosing the manuscript foods-2621886 entitled “Application potential of Trichoderma in the degradation of phenolic acid-modified chitosan” that we believe should be of strong interest to the general readership of the Foods journal.
We would like to note that in addition to addressing all reviewer’s valuable remarks, the authors placed additional editorial corrections including references to improve the quality of the manuscript. Below are our point-by-point responses to reviewer’s comments:
Reviewer #1:
In this manuscript the authors investigated the potential use of fungi of the genus Trichoderma for the degradation of phenolic acid-modified chitosan in compost. The results showed that the application of Trichoderma had a beneficial effect on the enzymatic activity in the compost.
The environmental pollution caused by plastics has been an important scientific and engineering issue. Thus, this is an important topic and the purpose of the manuscript is valuable. The paper is well written and clear, and need a minor revision.
The minor issues should be addressed:
- Line 334, “3.4. Fourier transform infrared spectroscopy – attenuated total reflectance.” I find it difficult to locate the peak positions described by the author in the figure. I suggest that the author annotate the figure with the functional groups corresponding to the peaks or indicate the peak values explicitly.
Thank you very much for this comment. We added peak values.
- 3600-2700 /cm. The notation of this unit is not standard and needs to be corrected. “3600-2700 cm-1”
Thank you for the comment. It has been corrected.
- Line 424. “Figure 3. The SEM images.” The figure caption describes Figure 3a, 3b, and 3c, but Figure 3a, 3b, and 3c are not included in the figure.
Thank you very much. We added Figure 3a, 3b, 3c.
- Line 102. “0.1M” should be “0.1 M”.
Thank you. It has been corrected
- Line 162-163. “with 4 cm−1 resolution, in the range 4000-600 cm-1”. The “-1” in the unit should be superscript.
Thank you. It has been corrected
- Line 329. “10 %” should be “10%”.
Thank you. It has been corrected
Reviewer 2 Report
Dear authors
My comments are ijn th attached file
All points should be carefully revised
Regards

Author Response
Dear All,
on behalf of myself and co-authors, I am enclosing the manuscript foods-2621886 entitled “Application potential of Trichoderma in the degradation of phenolic acid-modified chitosan” that we believe should be of strong interest to the general readership of the Foods journal.
We would like to note that in addition to addressing all reviewer’s valuable remarks, the authors placed additional editorial corrections including references to improve the quality of the manuscript. Below are our point-by-point responses to reviewer’s comments:
Reviewer #2:
Line 15 The abstract should be revised and be well develeoped.
Based on comments, the abstract has been corrected. Although I was careful not to duplicate the description of the results.
“Abstract: The aim of the study was to determine the potential use of fungi of the genus Trichoderma for the degradation of phenolic acid-modified chitosan in compost. At the same time, the enzymatic activity in the compost was checked after the application of a preparation containing the fungi Trichoderma suspension (spores concentration 105/mL). The Trichoderma strains were characterized by high lipase and aminopeptidase activity, chitinase, and β-1,3-glucanases. T. atroviride TN1 and T. citrinoviride TN3 metabolized the modified chitosan films best. Biodegradation of modified chitosan films by native microorganisms in the compost was significantly less effective than after the application of a formulation composed of Trichoderma TN1 and TN3. Bioaugmentation with a Trichoderma preparation had a significant effect on the activity of all enzymes in the compost. The highest oxygen consumption in the presence of chitosan with tannic acid film was found after the application of the consortium of these strains (861mg O2/kg after 21 days of incubation). Similarly, chitosan with gallic acid and chitosan with ferulic acid was found after the application of the consortium of these strains (849 mgO2/kg and 725 mg O2/kg after 21 days of incubation). The use of the Trichoderma consortium significantly increased the chitinase activity. Application of Trichoderma also offer many possibilities in sustainable agriculture. Trichoderma can not only degrade chitosan films, but also protect plants against fungal pathogens by synthesising chitinases and β-1,3 glucanases with antifungal properties.”
Line 16 I invite authors to add some numerical data.
The data were introduced in the abstract. I have not added the enzyme activity data in the abstract in order not to repeat the description of the results. After introducing data on the biodegradation of chitosan films, the abstract seems too long.
Line 17 Some technical approaches used in this investigation should be developed briefly
Thank you for the comment. It is now changed.
Line 24 A 2-3 concise and conclusive sentences should be added at the end of the abstract
Thank you for the comment. Three sentences of summary have been added.
Line 26 Insert keywords until 5
Keywords have been added.
Line 30 This sections should be revised, by checking English language
We appreciate this comment. All paper has been checked by a native speaker. Sweta Binod Kumar is a native speakerhe and she is also the co-author of the paper.
Line 35 I invite authors to use recent and proper references (2019-2023), and more sentences should be developed
Thank you for this comment. Reference were added:
“Priyadarshi, R., Rhim, J. W. Chitosan-based biodegradable functional films for food packaging applications. Innovative Food Science & Emerging Technologies; 2020, 62, 102346. https://doi.org/10.1016/j.ifset.2020.102346.
Wang, W., Xue, C., Mao, X. Chitosan: Structural modification, biological activity and application. International Journal of Biological Macromolecules; 2020, 164, 4532-4546. https://doi.org/10.1016/j.ijbiomac.2020.09.042.”
Line 36 L36-38 what is the link of this sentence, please be careful
Thank you for the comment. The sentence has been corrected according to the source. In this paper, the authors examined the mechanism of action of gallic (GA) and ferulic (FA) acids, a hydroxybenzoic acid and a hydroxycinnamic acid on Escherichia coli, Pseudomonas aeruginosa, Staphylococcus aureus, and Listeria monocytogenes.
Line 40 why authors focus this idea on the application of chitosan in food packaging ?
Thank you very much for this comment. The development of new biodegradable packaging materials is an alternative to current packaging that is difficult to degrade. Currently, food is packaged in materials containing various chemicals, e.g. bisphenols, which are harmful to humans. this is found in bottles and other plastic food packaging. Chitosan film, due to its fungicidal properties, can be used to package vegetables and fruit that are chemically protected against fungal growth.
line 48 introduce the inoculum strain of each fungi
Vivi et al. report inoculum of the fungus Chaetomium globosum ATCC 16021 in their experiment. We have added this information to the introduction. In contrast, Ohkawa et al. studied the biochemical oxygen-demand (BOD) biodegradation test of polyion complex (PIC) fibres, chitosan-gellan (CGF), and poly(L- lysine)-gellan, but they did not report fungal inoculum.
line 64 Authors should stress the novelty of this work
The novelty of this paper was showed in purpose of the research.
Line 67 please dvelop this sentence
The sentence has been changed.
line 68 some numerical data should be developed
In this paper, hydrophobins were detected by Westernblot, the authors do not provide data.
line 70 The objective was not clear, improve it
Thank you for this comment. The purpose of the research has been changed:
„Food packaging, which is produced on an industrial scale from materials that are difficult to degrade, represents a serious environmental problem. They take many years to decompose. An alternative to current packaging materials could be chitosan films modified with gallic, ferulic and tannic acids. These materials can be degraded in the environment by native microorganisms. However, not all native microorganisms are able to degrade chitosan films due to their biocidal properties. In order to accelerate biodegradation, a microbiological preparation based on Trichoderma has been developed to support their decomposition. Fungi of the Trichoderma genus are known especially for the production of antifungal enzymes, which may be an additional feature when applied to the environment.”
Line 70 how about the applicability of as strain in food industry?
The development of a microbiological preparation allows for their potential use by biotechnology companies producing preparations.
Line 101 How about the choice of these 3 phenolic acids
Tannic acid, ferulic acid, gallic acid
Thank you very much for this comment. Those 2 phenolic acids were selected based on our previous research where they were used successfully as cross-linker of biopolymers (J. Func. Biomater. 14, 2023, 69). These materials can be degraded in the environment by native microorganisms. However, not all native microorganisms are able to degrade chitosan films due to their biocidal properties. In order to accelerate biodegradation, a microbiological preparation based on Trichoderma has been developed to support their decomposition.
Line 105 this sentence needed a reference, how about the choice of this concentration?
Thank you for the comment. We prepare films by the same method as previously as it was recognised by us as effective method to obtain thin solid films.
Line 107 how about the chitosan preparation?
Thank you very much for the comment. We removed these sentences as chitosan was purchased and then used, without any modifications. We apologize for this misunderstanding.
Line 115 revise this format 105
It has been corrected.
line 131 authors are invited to concise some sentences
The sentences have been corrected.
line 163 how about the studied samples. Please describe this experiment
The part with line 163 is part of FTIR-ATR analysis. It is corrected now.
Line 231 all data should be linked since several data were provided in this MS. Proper tool should be used as Pearson, pca, hca
We have organized the information about the statistical tools used to clearly indicate which test was applied to which data.
Line 240 If author can establish a phylogentic trees xith proper standardised strains (TCC, NRRL, NBRC, ...)
The phylogenetic tree has been added .
line 265 this error was at 50%, please check it!!!!
We appreciate this comment, however, we had problem to respond. As indicated in the table's title, the values presented within it represent the mean and standard deviation. You mentioned a 50% error for the value 1.9 ± 0.01. However, the standard deviation value (0.01) accounts for only 0.5% of the mean value (1.9), not 50%.
Line 284 Italic form
It has been corrected.
line 288 some general data should be deleted, please be concise
The sentence has been changed.
line 312 at each level?
Thank you for this comment. BOD was after 8 days.
Line 390 pleasqe add supercripts in this tbale each mean should compared for each parameter at each concentration
Thank you for the comment. It is now done.
Line 401 this paragraph was ot clear, should be well written with proper references
Thank you for this comment. We added information that first sentence refers to Figure 2.
Line 426 Please improve this figure, the change should be clear between samples
Thank you for this comment. Figure 3 presents SEM images obtained for tested films. There are no significant changes as it is presented and discussed.
Line 530 please link all data with a proper statistical tool
Thank you for the comment. Its is now corrected.
Line 570 this part was short, please improve it with taken all suggested remarks
This part was changed
„Chitosan is known primarily for its antifungal properties. Chitosan-based liquid preparations have been developed for plant protection. Chitosan films are also used in medicine as a dressing material. The introduction of biocidal phenolic acids into chitosan film makes it possible to use it as food packaging to protect against the growth of both bacteria and fungi. The potential of modified chitosan films may be seen in the protection of fruits and vegetables. Such foil can protect them against the development of fungi. However, packaging materials should be degradable in the environment. Studies have shown that chitosan modified with phenolic acid is less readily degraded by native microorganisms. Only after the application of Trichoderma does the degradation of chitosan films occur more effectively. At the same time, studies have shown a beneficial effect of the use of Trichoderma on the enzymatic activity of compost. Chitinases, the activity of which increased after the introduction of the preparation, can control the development of phytopathogens”.
Round 2
Reviewer 2 Report
all points were revised,
Therefore, this current MS accept as it is
Best regards